# Zero-Trust Model for Smart Manufacturing Industry

**Biplob Paul [1] and Muzaffar Rao [1,2,3,*]**

1 Department of Electronic and Computer Engineering, University of Limerick, V94 T9PX Limerick, Ireland
2 Confirm–SFI Centre for Smart Manufacturing, Park Point, Dublin Rd, Castletroy, Co., V94 C928 Limerick, Ireland
3 Lero–Science Foundation Ireland Research Centre for Software, Co., V94 NYD3 Limerick, Ireland
* Correspondence: muzaffar.rao@ul.ie

**Abstract:** Traditional security architectures use a perimeter-based security model where everything internal to the corporate network is trusted by default. This type of architecture was designed to protect static servers and endpoints; however, we need to adapt to emerging technologies where serverless applications are running on containers, mobile endpoints, IoT, and cyber-physical systems. Since the beginning of the fourth industrial revolution (Industry 4.0), there has been a massive investment in smart manufacturing which responds in real-time to the supply chain and connects the digital and physical environments using IoT, cloud computing, and data analytics. The zero-trust security model is a concept of implementing cybersecurity techniques considering all networks and hosts to be hostile irrespective of their location. Over the past few years, this model has proven to be a remarkably effective security solution in conventional networks and devices. In this paper, the zero-trust approach will be fully explored and documented explaining its principles, architecture, and implementation procedure. It will also include a background of the smart manufacturing industry and a review of the existing cyber security solutions followed by a proposed design of the zero-trust model along with all the enabling factors for on-premises and cloud-hosted infrastructure. Various security solutions such as micro-segmentation of the industrial network, device discovery, and compliance management tools that are essential in achieving complete zero-trust security are considered in the proposed architecture.

**Keywords:** cybersecurity; zero trust; network security; access control; smart manufacturing; cyber-physical system; Industry 4.0; internet of things; cloud computing





## 1. Introduction

Zero trust is a strategic approach to cybersecurity that secures an organization by eliminating implicit trust and continuously validating every stage of digital interaction [1]. There is no generic definition of zero-trust, but it is seen as a security model which considers all sensitive data are not necessarily protected by firewalls and as an access control approach, trust must be established every time removing any assumptions from past decisions [2].

Most of the business applications and resources nowadays reside outside of the traditional perimeter. There are various options for organizations to choose from depending on their business needs and infrastructure. This migration from on-premises to public cloud or hybrid cloud environments results in more challenging cybersecurity issues, most of which are related to user access. The Zero-trust security approach is about eliminating the implicit trust between users, applications, and infrastructures within or outside of an organization's network. It is a methodology that considers the entire ecosystem of controls rather than just focusing on the narrow technology [1].

In 2011, the German federal government announced Smart Manufacturing, also known as Industry 4.0 to be a key initiative in shaping the future of the manufacturing industry [3]. Smart manufacturing focuses on the end-to-end digitization of all physical assets and integration into digital ecosystems with value chain partners. Ecosystems form when

a collaborative network of stakeholders, enabled by digital technology, come together in meaningful ways to meet shared objectives and solve shared challenges. The three key components of Smart Manufacturing are the Internet of Things (IoT), Cyber-Physical Systems (CPS), and Smart Factories.

Digital ecosystems can only function efficiently if all parties involved can trust in the security of their data and communication, as well as the protection of their intellectual property. Protecting the company and ensuring digital trust requires significant investment and clear guidelines for data integrity and security [4]. Even today cybersecurity is considered one of the characteristics of smart manufacturing technology rather than it being an integral part of the design. The denial-of-service attack on Ukraine's power grid in 2015 [5] and the spear-phishing attack on Germany's steel factory in 2014 [6] are a couple of examples that reflect this misconception. The severity of attacks on smart manufacturing systems is highly catastrophic because the potential outcomes of these attacks not only damage the production line or quality of the products but can also range from economic breakdown to injury and loss of human life up to an extent of nationwide effects [7]. Designing a sustainable security model such as zero-trust for the future smart manufacturing systems will bring down the level of cyberattacks thus lowering the severity of the damage.

Zero-trust model brings maximum security with some disadvantages like complexity, more manpower, slow down application performance, more cost and hamper productivity. These disadvantages and their mitigation are given in [8]. Zero-trust is not only securing traditional computing assets but has extended to the cloud and IoT spaces such as the one used for smart manufacturing alongside big data, cloud computing, AI, and machine learning [9]. The objective of this paper is to propose a zero-trust model for the smart manufacturing system which can secure the vulnerable communication channels between different manufacturing equipment such as the execution system and control system, the data that are sent to the cloud for analysis & intelligent control, and the admin access to these resources.

It is important to understand that zero-trust is not just one security product that can be placed in the infrastructure and confirmed to be secured. It is rather a concept that involves multiple aspects of closing down the holes in an environment with the use of different security products. The methodology of the proposed zero-trust architecture is based on the idea of using various security solutions that are essential in achieving complete zero-trust security. Solutions considered in the proposed architecture are listed below (details given in Section 7):

- The authentication process consists of signature validation by an asymmetric cryptographic algorithm;
- The authority of the operator is validated by comparing its identity with the predefined set of policies;
- Any implicit trust is removed between two communicating manufacturing devices;
- Every manufacturing phase is isolated from the others;
- Ideally, one of the systems in each production phase is responsible for uploading the manufacturing data to the storage server;
- In case of the involvement of clouding systems, an additional component called a cloud connector must be implemented in the private manufacturing environment;
- Communication channels are encrypted/secured with a proper authentication mechanism in place using certificates.

In short, a zero-trust model can be implemented by combining all the different security solutions to protect the smart manufacturing environment. The main contribution of this work is to discuss and propose the zero-trust architecture for a smart manufacturing environment. Traditional perimeter-based security is still being used in manufacturing industries, which is based on the concept that everything inside the firewall is considered trusted.

The rest of the paper is organized as follows: a zero-trust overview is given in Section 2, zero-trust architecture implementation is discussed in Section 3, security challenges for



smart manufacturing are highlighted in Section 4, existing cybersecurity solution for smart manufacturing is given in Section 5, Smart Manufacturing infrastructure without Zero-trust security is discussed in Section 6, security considerations and proposed security solution is given in Section 7, and Section 8 concludes.

## 2. Zero-Trust Overview

### 2.1. Principles of Zero Trust

There is no single reason for implementing zero trust, it is considered an improvement over the traditional perimeter-based security model which operates on the concept of trust [2]. Every user, device, and network traffic inside the border is trusted by default and this gives rise to easy lateral access to the attackers once they successfully cross the border, whereas in zero-trust everything is considered hostile, and trust must be gained through multiple parameters including (but not limited to) user authentication, authorization, verification of devices and services [10]. The main principles of zero-trust include:

- All critical data, assets, and services (DAAS) must be secured irrespective of their location (inside or outside of the corporate network) [10].
- No components are trusted by default and access communications must be encrypted at all times, even for channels within the intranet [11].
- Just-in-time access (JITA), where the authentication and authorization verification are done based on the set access policy exactly at the time of access request and will be valid for one session only [2].
- Just enough access (JEA) which corresponds to least privilege access; should be applied such that only enough amount of access required to carry out the task is provided and no more [2].
- Access control policies must be configured based on the data received from a maximum number of sources such as device health, type of resource accessed, etc. [11].
- Access is never granted based on history and must be evaluated every time it is requested [11].

### 2.2. Zero-Trust Architecture

The first rule of thumb while architecting the zero-trust model is that the requestor never has direct access to the resource [11]. As shown in Figure 1, all communications will go through the policy enforcement point (PEP) and even before the request reaches the PEP, the requestor is authenticated using a single or multi-factor authentication process depending on the environment. Upon successful authentication, the request is validated at the policy decision point (PDP) against the configured policies which are dynamic and can change based on the information provided by the policy authority. At this point, a decision is made by PDP whether to allow or deny access to the resource [2,11]. Apart from the policy decision and enforcement points, there may be other subsystems included supporting fully featured zero-trust architecture such as public-key infrastructure, logging, and analytics [2]. In Forrester's "The Zero-Trust eXtended (ZTX) ecosystem" [12], the criticality of enabling automation and orchestration is discussed explaining the extended capabilities of access control which makes it more dynamic based on the feeds received from visibility and analytic tools that tracks the activities in the network.

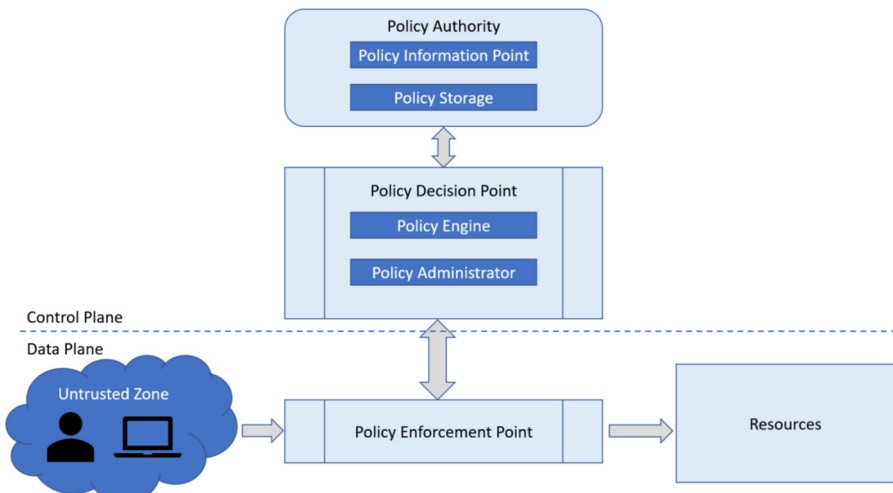

**Figure 1.** Zero-trust high-level architecture.

*2.3. Zero-Trust Network Access (ZTNA)*

Google implemented a zero-trust framework for its internal network called Beyond-Corp in 2015 [13]. This implementation includes a Device Inventory Database, which uses a concept called "Managed Device" and only these devices are allowed to access the enterprise resources. Device certificates are issued during enrolment to uniquely identify them in the database and are also used as one of the factors in the authentication. Beyond-Corp also implemented a Single Sign-On system for authentication, which issued a session token for the access of a specific resource. An internet-facing proxy in their network is also included, which delegates requests from internal and external clients to the appropriate back-end resource. All the public DNS registrations of Google's applications point to this access proxy.

Based on the concept of service-initiated architecture by BeyondCorp, Gartner 2019 introduced Zero-Trust Network Access (ZTNA) in [14]. According to this, an additional component called Connector is implemented in the internally protected network to accept incoming requests from users or devices after authentication. The connector then processes the request to verify access and connects the requestor to the service. ZTNA is rapidly replacing VPN-based access to critical services and will continue to phase out 60% of VPN usage by 2023 as per Gartner's prediction [12].

**3. Zero-Trust Architecture Implementation**

In Section 2, the principles and high-level architecture of the zero-trust model were discussed. In this section, a detailed explanation of zero-trust architecture and its implementation methods is provided.

*3.1. Data Access Scenarios*

As per the guide for zero-trust implementation by NIST [15], scenarios for data access in an IT environment are given below, and all of these must be secured after implementing the zero-trust architecture.

3.1.1. Employee Access to Corporate Resources

This particular scenario includes all the different possibilities of how an employee can securely access corporate resources such as payroll information, time-clocking systems, emails, etc. from any location. The corporate resource in this scenario can be either located on-premises hosted in a private data center or it can also be on the cloud managed by an external cloud provider.

### 3.1.2. Employee Access to Internet Resources

This scenario captures all the events where an employee makes any kind of access requests to the internet from within the corporate network or from outside the corporate network using an enterprise-managed device. The zero-trust architecture will be controlling such accesses, for example, any kind of social media access from an enterprise-managed device is restricted, and other internet resources when permitted, the traffic is not routed via the corporate network.

### 3.1.3. Third-Party Vendor or Contractor Access to Corporate Resources

In this scenario, all the access requests made by third-party vendors or contractors are captured. The requested corporate resource may be on-premises or on the cloud and access to them will be controlled and restricted by the zero-trust architecture based on the particular job function, the contractor is assigned.

### 3.1.4. Server-to-Server Communication

Servers in the enterprise often communicate among themselves, for example, a web server communicates with the application server, which in turn communicates with the database server for data retrieval, etc. These servers may all reside on-premises within the corporate network boundary or on the cloud and their communication channels may be internal or may even be through the internet in the scenario of a hybrid infrastructure.

### 3.1.5. Federated Integration between Enterprises

This scenario includes all the access requests made from the employees of one organization to the resources of another organization. This occurs in the event of business collaboration between two or multiple enterprises and a section of resources is exposed to be accessed by a certain group or sub-section of employees from another federated partner organization. The zero-trust architecture implementation will manage such types of access requests securely.

### 3.2. Dynamic Policy Implementation Scheme

As described in Figure 1, one of the crucial components of zero-trust architecture is its policy engine which controls access to resources based on dynamic policies. Every enterprise has some form of monitoring system and also SIEM (Security Information and Event Management) which collects the analytics and security events from multiple sources such as Firewalls, IDS, IPS, endpoint security, antivirus agents, etc., and analyses them to detect abnormal behavior [16]. Based on this abnormality and its criticality, a risk score is assigned to the users and/or to their devices. Table 1 shows an example of a series of suspicious events detected in SIEM and the corresponding risk scores assigned to each of them.

**Table 1.** Example of risk score based on security events.

| Events | Risk Score |
| --- | --- |
| Malware detected on the device | +10 |
| Connect to the corporate network remotely from an unknown location | +5 |
| Access an unusual internet site that has been reported as malicious | +5 |
| Execution of a file downloaded from the internet that deletes shadow copies | +20 |
| Account switch from user to a service account | +40 |

These risk scores for each user and device are fed into the policy engine to decide whether to allow or deny an access request for a resource. Depending on the criticality and sensitivity of the resource, there can be a threshold set for the risk score, beyond which access to that resource will be denied by the policy engine. For example, if a resource is set to bare a maximum risk score of 50 to be able to access, then the user with the above series

of security events will be denied access to this resource because the total risk score is, e.g., 80, which is higher than the threshold for the requested resource.

Another classification of policy is based on the job function of the requester. The level of access granted to the resource will be determined by the role of the requester and is also known as RBAC (Role Based Access Control). Each enterprise resource is assigned different types of access roles, for example, an incident management application will have an incident reporter role, a service requester role, a technical support role, an auditor role, a report manager role, and an administrator role. Now, a user who just wants to report an incident should not be able to view/modify incidents reported by other users. Similarly, a reporting manager is only responsible for generating reports from the application and should not have access to create a user. Whereas an administrator role will have complete access to the application. The policy engine validates the user account and authorizes the least privileged access to the resource depending on the user's role. This can also be tied up with the risk score assessment such that a user with a moderate risk score device connecting to an application will get less privileged access than what their account is entitled to rather than completely denying the access.

### 3.3. Policy Evaluation Process

After the policies are set in the zero-trust architecture, the next step is the evaluation process. There are multiple scenarios of the policy evaluation, which depend on the infrastructure of the organization. For an on-premises architecture, where the enterprise resources are hosted within the organization's data center internally, the request for the resource will be evaluated by the policy enforcement point before the request reaches the endpoint. Figure 2 demonstrates such architecture where the SIEM system collects all the analytics and security events from multiple sources to decide upon a risk score before feeding them into the policy decision point which is been queried continuously by the policy enforcement point as and when it gets a request for access. Depending on the decision made by all the policy combinations, the access request is either granted, denied, or granted with lower privileges.

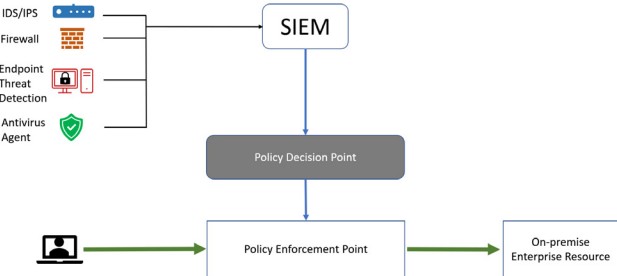

**Figure 2.** Policy evaluation process for On-premises infrastructure.

Another scenario is when the enterprise resources reside on cloud infrastructure managed by a third-party cloud provider. Here, the user accesses the resource directly through the internet because they are on the cloud, and all the resources are configured to contact the policy decision point when there is an access request. The policy decision point evaluates the policies to make a decision before responding to the cloud resource which then allows or denies access to the requester. In a complete cloud environment, the policy decision point and the SIEM system also reside in the cloud reducing the communication time. Figure 3 shows an example of such architecture.

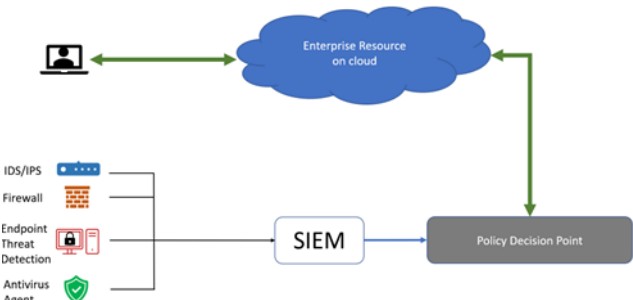

**Figure 3.** Policy evaluation process for cloud-hosted enterprise resource.

A public cloud or even a hybrid cloud environment has blurred the physical boundaries of a corporate infrastructure. The applications are no longer confined within the firewall-protected zones and most of them are migrated to the cloud as a SaaS (Software-as-a-Service) based or an organization-hosted application residing on a cloud infrastructure that is owned and managed by third-party cloud providers. These cannot be protected by traditional perimeter-based security anymore and needs an elevated solution such as the zero-trust model to completely secure their accesses.

## 4. Security Challenges of Smart Manufacturing

Smart manufacturing systems aimed to bring in effective solutions to achieve high-quality products and be flexible about the approach which gave rise to one of the significant security risks because of it being seen as a secondary concern rather than an essential integral part of the development and deployment process. Figure 4 displays all the characteristics, design principles, and enabling factors of smart manufacturing and it can be seen that cybersecurity is defined among the characteristics rather than being one of the design principles. Security cannot be just added to a system as one of its characteristics, it needs to be integrated during or even before the design and must be a continuous process.

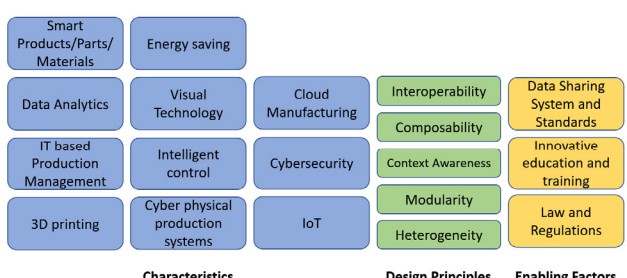

**Figure 4.** Logical components of Smart Manufacturing.

When compared to the capabilities of conventional manufacturing processes, the level of technical sophistication, integration, and automation are far beyond in smart manufacturing systems. The previous concept of computer-integrated manufacturing had more of a master/slave architecture, in which communication was typically initiated by the master, and securing such an environment was easier when compared to a decentralized architecture of industry 4.0. Here, all products and machines are autonomous and will actively participate in the production line. An open environment is vulnerable to a wide range of active and passive attacks such as DDoS (Distributed Denial of Service), MITM (Man-in-the-Middle), and Eavesdropping. AI and machine learning are essential components of industry 4.0 and all the data from each autonomous equipment are uploaded to the cloud for future learning and analytics. Poisoning of such data that are sent for analysis and intelligent control could have a significant effect on the physical integrity and quality of the output. Given the high dimensionality of such data, a minor change could impact high on the learning methods and could even stay undetected [7].

## 5. Existing Cybersecurity Solutions for Smart Manufacturing

There are three base levels of approaches that can assure the security of industrial control systems (ICS) [17]:

- Separating the industrial network from the corporate network and hardening the perimeter with firewalls and DMZ. Restricting internet access to the ICS systems and closely monitoring their interactions with other devices.
- Defense-in-depth implementation in the network to contain any breach from spreading across the entire industrial network. There may be a need for segmentation and micro-segmentation with a layer of defense between each.
- Isolating remote access to a dedicated network segment with the least privilege and audit control mechanism. Modern remote access solutions provide a more granular level of control such as which endpoints and applications can be accessed remotely, by which users, and during what time period.

Let us have a look at a few existing cybersecurity measures for smart manufacturing.

### 5.1. Cybersecurity Countermeasures

### 5.1.1. Cryptographic Techniques

One of the most common and widely used cybersecurity countermeasures is encryption which is available at multiple levels. A smart manufacturing environment consists of multiple devices and equipment. Hence, there are also networking and software protocols for communication among them and with the human operator. Cryptographic techniques such as symmetric encryption, hash functions, digital signature, key agreement, and distribution protocols are used in these types of communications to achieve confidentiality and integrity of the data which also ensures that only authorized entities can have access to it. As discussed in [18], there are several key management techniques. The key distribution center (KDC) is one of the key management solutions where the KDC server will generate and distribute the session keys to the communication entities. Another solution for key management is using point-to-point architecture, where the two comminating parties will either agree on a key exchanging random nonce, or one of the entities will generate the key and distribute it to the other entity encrypting it with a long-term symmetric key. In the scenario of asymmetric keys, a standard PKI can be one of the key management solutions which are responsible for certifying the public keys of each entity by digital signature. They can also verify and revoke the certificates if necessary.

### 5.1.2. Intrusion Detection Systems

Vulnerabilities are inevitable regardless of the layers of defense in the system. To avoid this, it is necessary to set a standard behavior of the system and detect any abnormality. IDS are classified into two categories, host-based or network-based and signature-based or anomaly-based. In the cases of smart manufacturing, host-based IDS implementation is quite impractical, as the IoT devices are incapable of hosting an IDS mechanism within the equipment, because of their compute-intensive nature. Whereas a network-based intrusion detection system, being a separate standalone device can capture the evidence from the whole segment of the network in which it is placed.

Signature-based IDS first creates a standard, based on the knowledge of previous attacks or known vulnerabilities and then compares the traffic activities to identify an intrusion. In the case of anomaly-based detection techniques, the IDS looks into the network traffic for any deviation from the standard set of expected system behavior. In the case of the manufacturing industry, setting a standard behavior is convenient by modeling the system operation which does not change very often. In the scenarios where the equipment transactions are modified by the supply chain, there can be a machine learning technique implemented for the anomaly-based IDS.

### 5.1.3. Security Training and Incident Management

Security of the systems is incomplete without skilled system users who are trained for the security of equipment against attacks such as social engineering and phishing. Periodic training sessions and awareness programs must be conducted, especially for the personnel who uses and manages the manufacturing systems. Although there are countermeasures, successful attacks are inescapable and when it happens, it is pivotal to be able to handle the incident by stopping any further spread of the attack and bringing the system back into operation as soon as possible. As mentioned previously, the consequences of attacks in smart manufacturing can be fatal causing damage to the environment and people. Thus, it is essential to have the required backup resources available before the attacks take place.

### 5.2. Security with Software-Defined Networking (SDN)

The manufacturing execution system (MES) sits in between the ICS and front-end applications used by human operators. The data collected by the sensors are stored, analyzed, and compared to determine the performance of the machine and the quality of the product coming out of the manufacturing process. The intercommunication between these systems makes the programable logic controllers (PLC) vulnerable to attacks causing hindrance in production and exposing other connected systems such as MES. Attacks, based on network scanning and probing are the most usual ones for MES and a distributed network architecture consisting of multiple segments guarded by firewalls is an effective countermeasure for such kinds of attacks.

In these types of networking environments, it is necessary to construct secure and flexible access control rules which give rise to the need for software-defined networking (SDN). SDN is an architecture where the forwarding function (data plane) of the networking devices is decoupled from their network control function (control plane) making it centrally manageable programmatically [19]. The SDN architecture allows network administrators to dynamically control the traffic flow by automated SDN programs which are independent of the underlying software of the networking devices. This approach greatly minimizes the threats in ICS networks as they can be accessed and controlled on demand. Figure 5 shows the network structure of the SDN firewall designed for smart manufacturing systems [20]. In this architecture, the SDN firewall consists of two critical components, the packet filtering function which allows or denies access to the ICS based on the access control rules, and the firewall controller stores the configuration of the manufacturing system and the rules created and managed by the administrators.

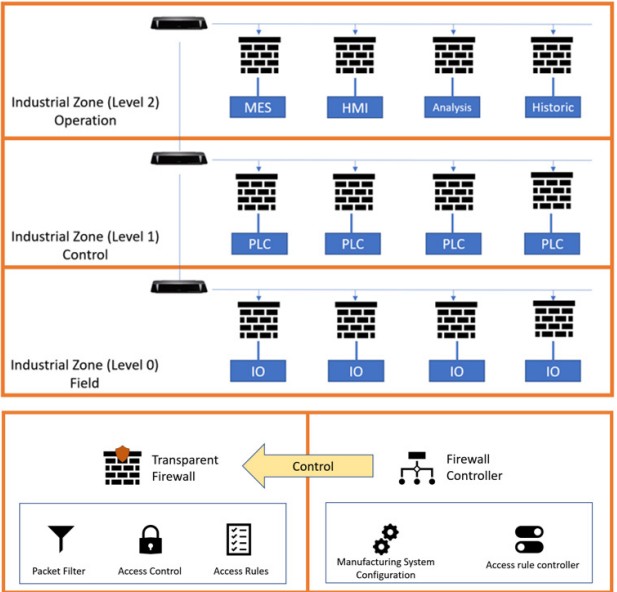

**Figure 5.** SDN Firewall for Smart Manufacturing.

### 5.3. Artificial Neural Network for Threat Detection

There are algorithms that can detect cyberattacks by analyzing the data collected from the events that occur in manufacturing environments. However, the main concern is the accuracy of those algorithms which can lead to false alarms or undetected intrusion. Computational Intelligence Systems (CIS) are flexible decision-making systems that are capable of handling a large amount of unstructured data in their decision-making process. It uses techniques such as Machine Learning (ML), Artificial Intelligence (AI), and neural networks which makes them suitable for designing the algorithms for detection systems. One such algorithm is defined in [21] which is designed using three components: neural network, genetic algorithm, and Neural Network Oracle (NNO). Figure 6 demonstrates the framework for the NNO algorithm concerning cyberattacks in smart manufacturing systems.

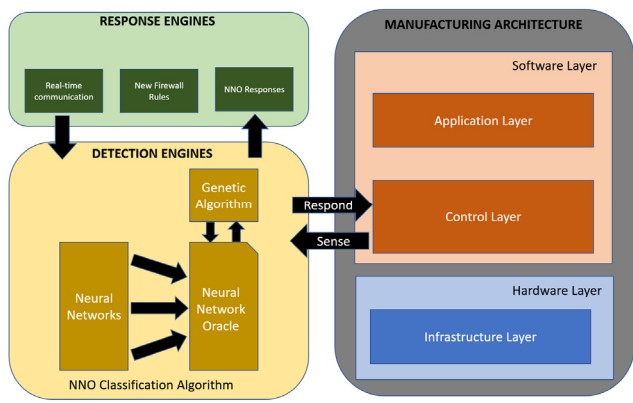

**Figure 6.** NNO Classification algorithm framework.

Artificial Neural Networks (ANN) work in the principle of biological processes to solve AI problems by self-learning based on training. The first set of audit data will be trained by the group of neural networks and the output is sent to NNO. The Genetic Algorithm (GA) which works similarly to the biological evolution process is responsible for defining the most accurate values to reduce the error in the detection algorithm. The NNO uses these values provided by the GA and the dataset received from the neural networks to increase the accuracy of the detections.

### 5.4. Blockchain Security in Smart Manufacturing

Blockchain works on the principle of the linked list structure combined with cryptographic techniques. Each data block holds the timestamp and a link to the previous block which is actually the cryptographic hash of the data in the previous block, forming a chain represented as a Merkle Tree. Since each block in the blockchain contains a fingerprint of data in the previous block, it is impossible to make any changes in the data without being traced unless the modification is carried out in all the subsequent blocks. The benefits of implementing blockchain in securing the smart manufacturing environment include resistance against data tampering, enhanced system resilience, and immutable manufacturing [22].

The data security categories, confidentiality, integrity, and availability could be strengthened by using blockchain. In the case of confidentiality, the distributed and cloud-based environment of smart manufacturing makes access control more complex. However, with blockchain, the public-key system variant and smart contracts provide privacy and dynamic access control policies for data modification. The confidentiality for transactions is provided by Zero-knowledge proof and blockchain, based on homomorphic encryption which enables calculations on the encrypted data without revealing the actual information. In the case of integrity, the chain structure of the blockchain along with its signature verification technique safeguards the data and transactions against any integrity loss. In order to achieve data accountability blockchain is used to assemble the provenance records across

the smart manufacturing distributed environment. In terms of availability, it is hard to detect fault quickly in a distributed manufacturing environment because it can cause by a single node and due to the decentralized architecture, it may remain undetected. However, with blockchain, there exists a replica node for each block of data that holds a copy of it and thus provides availability. Being a secure approach to smart manufacturing, blockchain implementation encounters many technical challenges which include a practical consensus algorithm, privacy protection mechanisms, and mutability of data. It is also difficult to balance the cost of implementation, the complexity of the system, and managing security. Moreover, due to the increasing size and number of data blocks, the analysis capability of blockchain is inefficient.

### 5.5. Other Security Solutions

### 5.5.1. Solutions by Security Firms

Two organizations, IBM and ABB have collaborated in 2020 to focus on the cybersecurity of operation technology and developed a security event monitoring service for smart industrial operations [23]. This is a combination of the process control system provided by ABB and the security event monitoring platform of IBM known as QRadar. They jointly developed a reference architecture that is designed to react to the security incidents of complex industrial environments. As a result of this collaboration, the process control system collects event log details and shares them with IBM QRader which identifies security anomalies and threats using Artificial Intelligence (AI). This reduces the number of security events causing disruption in production which leads to financial loss to the manufacturing company.

In 2019, Deloitte and the Manufacturers Alliance for Productivity and Innovation (MAPI) have produced guidelines to build effective manufacturing security programs to identify, protect, respond, and recover from cyberattacks on smart manufacturing industries [24]. As per them, the operational technology (OT) organizations should invest in cybersecurity maturity assessment, establishing security governance programs for ICS, Identifying and prioritizing actions based on the assessment results, and finally building security. Important aspects to consider while building security controls are network segmentation models, passive monitoring solutions, securing remote access to the industrial network, privileged access management, and backup resources for fault recovery.

### 5.5.2. Honeypot-Based Solutions

Honeypot systems are similar to IDS which detects and alerts in the event of any infrastructure breach. However, the data or information that resides within honeypot systems are fake but seems legitimate and valuable resource that will attract the attackers. This is a reverse phishing technique where an isolated and monitored environment is created with some form of meaningful and valuable looking resources to trap the attackers. The advantages of creating honeypot systems include awareness of the threat of being attacked by someone, while the attacker is investing time in attacking the honeypot system, other valuable resources can be alerted and secured, and finally, it gives more time for the security incident response team to react and stop the attack [21].

### 5.5.3. Digital Twin-Based Solutions

A digital twin system is a real-time synchronized clone of the actual infrastructure. In the scenario of the manufacturing industry, a digital twin is a virtualized representation of the physical manufacturing objects that provides a possibility to visualize, monitor, and predict the state of such systems. The synchronization of the updates between the cyber-physical system throughout its life cycle and its digital twin is provided by a digital thread. The purpose of building a digital twin is to simulate a safe breach to analyze the system behavior under attack and estimate the potential damage. The result of this analysis will facilitate the security and safety design mechanism to produce a more robust and fault-tolerant architecture [21]. Digital twin systems can be used for penetration testing to

identify vulnerabilities in the infrastructure and any unnecessary functionality of a device can be revealed. They also provide testbeds for security tests to be done virtually and fix vulnerabilities without impacting the real systems.

Despite all the security challenges addressed by the digital twin architecture, they also pose a threat that allows attackers to exploit the digital twin systems to launch an attack on the actual production systems. Once an attacker breaches the perimeter of digital twin infrastructure, even though there is no impact on the real systems, the attacker can gain knowledge of the network and system configurations as they are replicas of the underlying cyber-physical systems. The digital thread, which is the link between the original system and its digital twin is an attractive target for the attackers to either gain access to the manufacturing environment or manipulate the synchronization data to divert the system into an insecure state [25].

## 6. Smart Manufacturing Infrastructure without Zero-Trust Security

Currently, with all the available security solutions discussed previously, the smart manufacturing environment is still lacking an efficient access control mechanism and continues to rely on a perimeter-based security model with implicit trust between the components residing in the internal network. Figure 7 shows an overview of the communication between an operator and the manufacturing equipment. There is no clear segmentation between the industrial and corporate network and any device which is connected to the intranet is allowed by default to communicate with any manufacturing equipment directly.

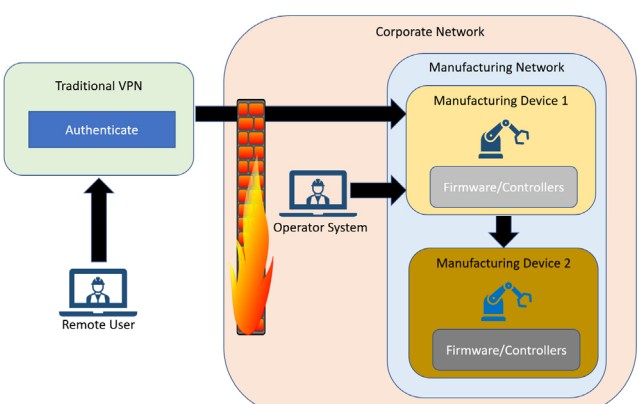

**Figure 7.** Current architecture of manufacturing industry without zero-trust.

In Figure 7, one operator is trying to connect to manufacturing device 1, from within the corporate network and they are allowed to do so without any authentication or authorization validation process since the source IP is internal and trying to connect to another IP in the same network. The other user, who is remote outside the corporate network will connect using a traditional VPN tunnel which uses a one-step basic authentication and once the connection is allowed to the corporate network, the remote user is free to access any resource or equipment without any further validation.

Similarly, the manufacturing equipment when communicating among themselves is allowed to do so without any authentication or validation. In general, this type of open access, trusting the internal devices by default is a threat to the smart industry. With an increased number of smart equipment such as IoT and CPS, there are several independent interactions between them without any involvement of a centralized server. Once a cyberattack penetrates the boundary, it is quite impossible to control the spreading of infection. Hence, monitoring each of these communication events and micro-segmentation of the network with a proper access control mechanism for every access will reduce the attack vector.

## 7. Security Considerations and Proposed Zero-Trust Security Model

Here, different aspects of potentially unsecured sections of the smart manufacturing industry will be discussed along with a proposal for securing them with zero-trust architecture. There are several kinds of data and communications encountered during the production cycle but only a few of them are critical for protection and a security solution should focus on them. The methodology used to create a zero-trust architecture is shown in Figure 8.

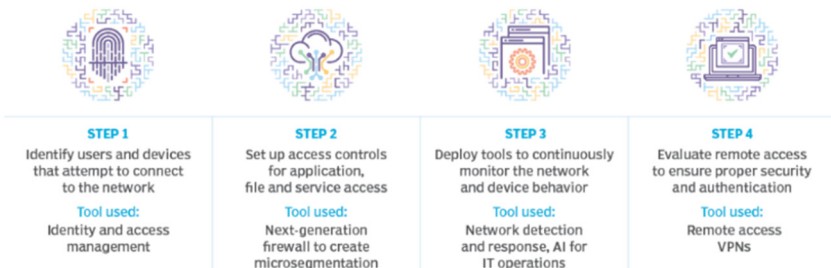

**Figure 8.** High-level steps involved in designing a zero-trust model for the manufacturing environment.

Four basic steps cover the complete design aspects of the zero-trust model. The first step is to identify the network-connected devices, step 2 involves the access management of all these devices to the manufacturing resources. Step 3 is to monitor the network and communications with the help of tools such as intrusion detection systems. The last and final step is to control the remote access to the OT environment smart manufacturing.

### 7.1. Security Considerations to Support Zero-Trust Model

7.1.1. Identity and Access Management for Operational Technology

The concept of Identity and Access Management (IAM) works in a similar way for the OT system as it does in the IT environment. The OT application owner creates Role-Based Access Control (RBAC) in the IAM, and the application users request access from the self-service tool which is verified and granted by the owner. It is critical to maintain separate accounts for IT and OT environments managed in different directories and set a user policy to keep a distinct username and password for them. Figure 9 below demonstrates the steps involved in implementing the access control process for OT applications.

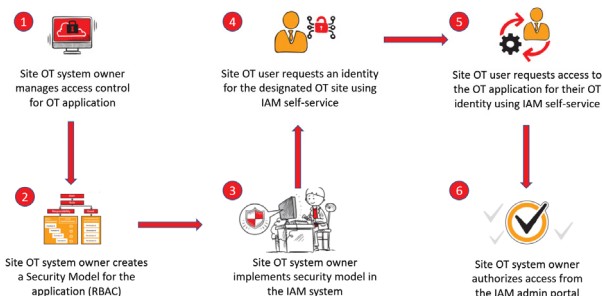

**Figure 9.** Identity & Access Management for OT applications.

The OT application owner creates a role-based access control policy for the particular application on the IAM admin portal. The OT application user first requests an identity from the IAM self-service portal and then requests access to the OT application for that newly created identity. The site owner authorizes and grants the least privileged access to the user based on their role in that application. This ensures that the right person has the right level of access to the OT applications and is also in compliance with termination and periodic access review processes in the event of user role change. User IDs for accessing OT applications are kept separate from the ones used for IT applications and also stored in different credential stores to impede malware propagation.

### 7.1.2. OT Network Segmentation via Firewall

Network segmentation of the OT environment from the IT network must be guarded by firewalls which will enable the system owners to exactly understand how and what data are transferred between the segments. One of the options to enable this is Isolated Computing Environment Firewalls. Figure 10 shows the working of such firewalled network.

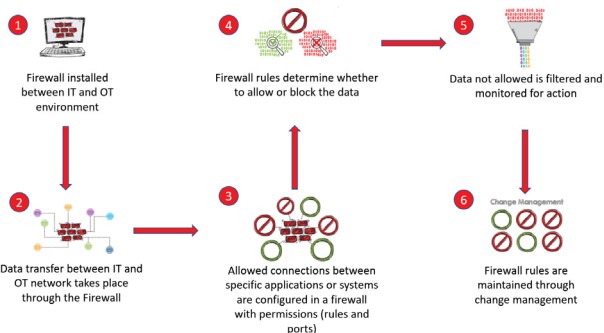

**Figure 10.** Network segmentation workflow.

After the installation of the firewall and configuring the rules by the administrator, all data transferred between the IT and OT network are controlled by the defined rules as to whether allow or deny the data. All the traffic that was filtered is then monitored for analysis to build foundational firewall rules. Always use implicit deny rules and monitor the policies for a certain period of time for example 90 days. Any change in firewall rules must go through a change management process. The benefit of separating IT and OT networks is to provide a barrier between them which will ensure increased resiliency against cyber threats passing from one isolated network to another.

### 7.1.3. Discovery of Network-Connected Devices in OT Environments

Enterprise Discovery tools are implemented for identifying every network-connected device or firewalled environment which will provide the capability to categorize them and also identify firewalls and unused subnets to reduce vulnerabilities. Figure 11 below shows the process flow of the device discovery. All subnets and VLANs are to be associated with a site code and create schedules for the discovery of network and approved OT subnet. This will enable the identification of devices, network addresses (IPs), and firewalled environments. The discovery result can then be utilized to create and update the visibility report for providing access to approved subnets/VLANs.

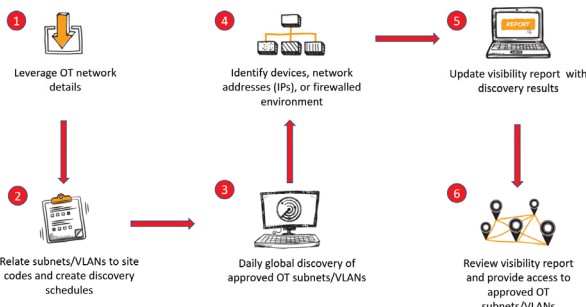

**Figure 11.** Device discovery technique in a firewalled environment.

The advantages include targeted incident and vulnerability response, providing awareness and input into other OT cybersecurity processes, and providing visibility into subnet/VLANs and all network-connected devices.

### 7.1.4. OT Endpoint Compliance Management

Endpoint management applications to be implemented that continuously scans network endpoints for vulnerable misconfigurations and compliance violations. The agent must be installed on all eligible OT endpoints and discoveries are made on the local OT network to identify other potential and manageable endpoints. The agent will collect data for OT assets, software, and patching inventory in real-time and then they will be classified as managed or unmanaged. Figure 12 shows how this will work and at the end of the process, administrators will be able to identify the endpoints that are out of compliance. Automation can be set up to take measures on those endpoints immediately to comply with organization standards and also automatically update the dynamic access control policies for these endpoints to be more restrictive until the time they are out of compliance.

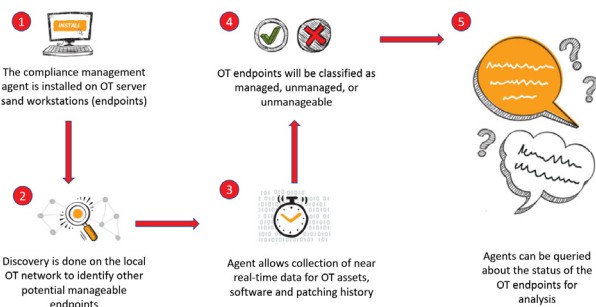

**Figure 12.** Endpoint compliance management overview.

The benefit of implementing a compliance management process is to have each site more secure through real-time insight into OT asset configurations, installed software, and missing patches. It also provides the detection of unmanaged OT assets and extensive insight into managed OT assets with reports for vulnerability assessment.

### 7.1.5. Privilege Remote Access

Remote access to the OT environment is not direct but through a Privilege Remote Access (PRA) jump server. All connections from the external network will be directed to the jump server via a VPN tunnel. This jump server will act as a proxy to the OT environment and all the connections are filtered through a set of firewall rules and sessions are automatically recorded for audit purposes. Each user connecting to the OT network from the internet must go through a series of authentication and authorization beginning with connecting to the VPN tunnel with MFA and then on the jump server, they must authenticate again to get connected to the OT environment.

Role-based and least privilege access is provided for remote logins and separate accounts must be maintained for IT and OT environments. Even if a user's IT account credentials get compromised, the malicious party could only connect through the VPN tunnel to gain access to the jump server but cannot go beyond that. Figure 13 demonstrates the flow of a remote connection.

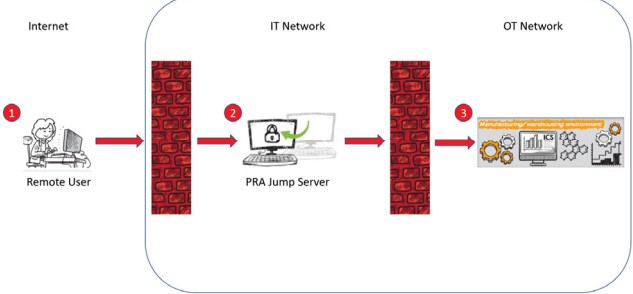

**Figure 13.** Privilege Remote Access management.

### 7.2. Proposed Zero-Trust Security Model

The proposed architecture is based on the previously discussed zero-trust architecture consisting of policy and security enforcement points. In Figure 14, two main components for authenticating and authorizing within the identity and access management platform sit between the operator's system and the manufacturing device. This eliminates any implicit trust between the operator and the machine. Two other important tools included are Enterprise Device Discovery System and Endpoint Compliance Management System. They both function independently to feed in the information collected from the user and device for efficient authentication and dynamic access control mechanism. The Enterprise Device Discovery System checks and validates the identity of each newly connected device into the network against the approved list of devices present in the database before assigning user certificates that are required for authentication with the IAM. On the other hand, Endpoint Compliance Management System monitors the device status regularly against the pre-defined compliance policies with the help of an agent installed on the system. These compliance policies can vary based on the business needs such as minimum OS version, patch version, antivirus status, password complexity, etc. Any device that is reported out of compliance will immediately get flagged in the policy decision point and access to resources will either be blocked or limited based on the severity of the violated compliance.

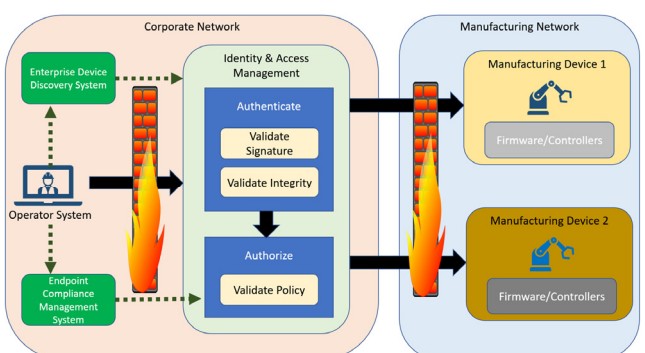

**Figure 14.** High-level architecture of Zero-Trust Security for Smart Manufacturing.

The authentication process consists of signature validation by an asymmetric cryptographic algorithm such as DSA (Digital Signature Algorithm) or RSA (Rivest, Shamir, and Adleman) where the public key is used to validate the authenticity of the data confirming that the sender is in possession of the appropriate private key. The DSA is preferred over RSA because of the speed of signature generation and validation [26]. The other component is integrity validation, which is done by a one-way hashing algorithm, SHA-3 (Secure Hash Algorithm-256) on the transmitted data creating a message digest of 256 bits and then encrypting it either with a symmetric algorithm such as AES in combination with a pre-shred key or by a public-key cryptographic algorithm. To make these processes of signature and integrity validation faster, they could be combined in one step where the hash of the information generated by SHA-3 is signed using DSA with the private key of the transmitter.

The next step is to validate the authority of the operator by comparing its identity with the pre-defined set of policies. The validation process consists of verifying the operator's location and the network they are connected to, verifying the operator's system and its security configurations such as antivirus and firewall status, and verifying the device and user certificates. The list of policies to be validated during each transaction may also contain dynamic policies which are based on the number of previous failed login attempts, system patch and OS update status, presence of any backlisted or absence of any whitelisted applications, etc. This two-step verification ensures instruction data integrity and the operator's identity and authority. Further, if there is a need for the operator to connect to the other manufacturing equipment, the same process is repeated instead of granting access based on the previous validation outcome.

Another consideration for a complete zero-trust architecture is removing any implicit trust between two communicating manufacturing devices. Considering a generic manufacturing environment, every phase of the production line will generate a certain identification for the batch of products manufactured in one cycle. This identity can be a product serial number or a batch number if there is a bulk of them produced in each cycle. There is always a simple communication from the ICS systems to the servers reporting each production cycle's batch number and product count along with the time and date of manufacturing. This is one of the many examples when the manufacturing systems will automatically send data to the server for audit control and business documentation. All these communications must be encrypted and authenticated every time the data is sent for each cycle of production even though the communication is internal to the OT network.

Every manufacturing phase is isolated from the others and no direct communication is possible between them to ensure if a system in one phase is under cyberattack, then it can be contained within that phase without being able to spread across other phases. The only common link between them is the storage server and thus it is very critical to secure the communications between the manufacturing devices and the storage server as shown in Figure 15.

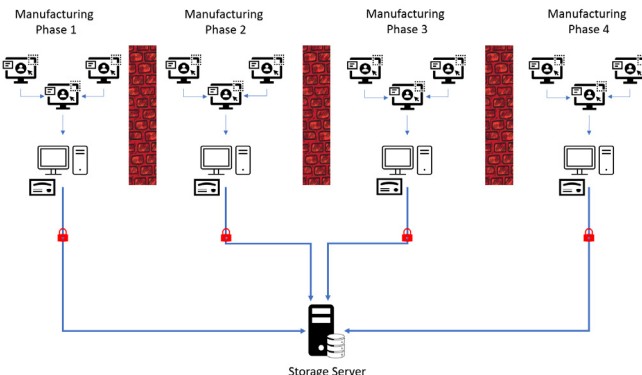

**Figure 15.** Zero-Trust security architecture without any implicit trust.

Ideally, one of the systems in each production phase is responsible for uploading the manufacturing data to the storage server, hence it is convenient and faster to use symmetric encryption, such as AES with a pre-shared key between the manufacturing device and the storage server for data encryption. The advantage of using the symmetric algorithm is it is faster when compared to an asymmetric one; however, the key management and its storage can be a challenge. Hence, the key management can either be handled by a centralized key distribution server (KDC) responsible for distributing session keys to all the manufacturing devices in every phase using the Kerberos protocol. However, in the case of limited infrastructure, keys can also be generated by one of the communicating entities and transported to the other using an asymmetric key distribution protocol. Certificate-based authentication of the identity for each manufacturing device while sending the data will provide data integrity and ensure that it was not modified during transmission. This can be achieved by generating digital certificates of public key signed by a trusted Certificate Authority (CA), and the process of a digital signature using DSA, as explained earlier in this section.

While the above architecture is best suited when the storage server is hosted on-premises inside the manufacturing network, the design will change in the scenario of cloud storage hosted and managed by a third-party cloud provider. In that case, an additional component called a cloud connector must be implemented in the private manufacturing environment which is responsible for maintaining a secure connection with the cloud and any data that are to be stored will go via this cloud connector component. Figure 16 below shows the architecture of a zero-trust implementation in a hybrid environment.

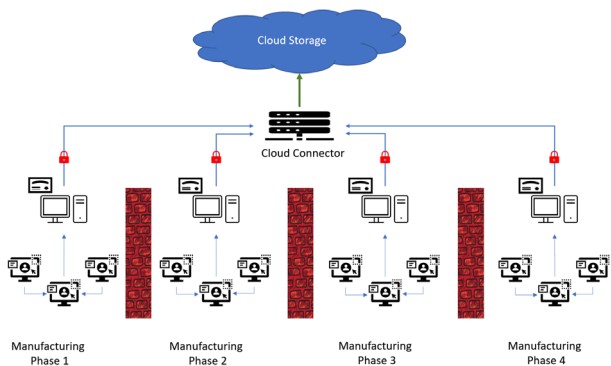

**Figure 16.** Zero-trust architecture for cloud-hosted data store.

All the communication channels are encrypted and secured with a proper authentication mechanism in place using certificates. Additionally, the cloud connector is responsible for policy evaluation before sending any data to the cloud. Since there is only a single point of contact for the cloud storage, it is configured in such a way that it only accepts requests from the cloud connector's IP address. The firewall rules are also set to support this configuration.

## 8. Conclusions

The zero-trust concept, since its inception, has already been a successful security model for IT infrastructure and many IT organizations are implementing this model from different vendors to secure their environment and manage their resource access in a controlled manner with the help of dynamic access control policies. With that said, the success of this solution in IT infrastructure can be considered as an example to motivate the security professionals in the smart manufacturing industry to implement the zero-trust security model. The manufacturing industries are still following the traditional perimeter-based security where everything inside the firewall is considered trusted and focuses on production with a very minimum priority on securing the industrial systems. The current security architecture for the manufacturing industries is good enough when there is no or very minimum communication between the industrial systems. However, the Industry 4.0 era has brought in many critical components such as machine learning, Artificial intelligence, cloud computing, etc. to automate manufacturing and business processes. Unfortunately, the more complex the environment is, the more communication between the components and thus, it is more susceptible to cyber-attacks.

With the introduction of cloud computing, most of the manufacturing resources have been migrated to the cloud reducing their management overhead. This type of hybrid architecture increases the workload of access management because communication between the private infrastructure and the cloud is through the internet. Any hole in the access policies will have a catastrophic impact when exploited by hackers. Although the zero-trust model brings in maximum security in terms of access validation, authentication, and authorization, the major drawback of this is the lack of proof for a successful implementation. There are currently multiple vendors in the market offering zero-trust security; however, it is extremely critical to understand that zero-trust is not just one security product that can be placed in the infrastructure and confirmed to be secured. It is rather a concept that involves multiple aspects of closing down the holes in an environment with the use of different security products such as firewalls, identity & access management platforms, secure remote access solutions, SIEM systems, endpoint security tools, and many others. Depending on the existing architecture of the industry, a solution needs to be designed considering all the possible threats. Only then a zero-trust model can be implemented by combining all the different security solutions to protect the smart manufacturing environment.

In the future, this research can be further explored by the addition of new and upcoming security technologies such as secure web access, VPN-less remote access, and cloud security when the industrial infrastructure starts migrating to the cloud.

**Author Contributions:** Conceptualization, B.P. and M.R.; methodology, B.P.; validation, B.P. and M.R.; formal analysis, B.P.; investigation, B.P. and M.R.; resources, M.R.; writing—original draft preparation, B.P.; writing—review and editing, M.R.; supervision, M.R.; project administration, M.R.; funding acquisition, M.R. All authors have read and agreed to the published version of the manuscript.

**Funding:** This work has received support from the Higher Education Authority (HEA) under the Human Capital Initiative-Pillar 3 project, Cyberskills.

**Institutional Review Board Statement:** Not applicable.

**Informed Consent Statement:** Not applicable.

**Data Availability Statement:** No new data were created or analyzed in this study. Data sharing is not applicable to this article.

**Acknowledgments:** Thank you to the Department of Electronic and Computer Engineering at the University of Limerick for supporting this work as part of the Masters in Information and Network Security program.

**Conflicts of Interest:** The authors declare no conflict of interest.

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
