# Peer review of "Zero-Trust Model for Smart Manufacturing Industry"

_applsci, doi:10.3390/app13010221_

Round 1

Reviewer 1 Report

The authors have reviewed zero-trust architecture in detail and its application in smart manufacturing industry. The literature and progress is up-to-date in the manuscript. However, authors should discuss some key limitations of zero-trust architecture and their mitigation. 

1. The biggest challenge with Zero Trust is that it can be complex to implement. The fact that every user, device, and application must be authenticated and authorized adds an extra layer of complexity, particularly for organizations with a large number of users or devices can be problematic.

2. One of the biggest complaints about Zero Trust is that it can slow down application performance. This is because every user, device, and application must be authenticated and authorized before accessing data or applications. Means latency will be the problem in these kinds of networks.

3. Another disadvantage of Zero Trust is that it can be costly to implement. This is because it requires more manpower and additional security measures, such as multi-factor authentication that can add to the overall cost of the system.

4. The final disadvantage of Zero Trust is that it can sometimes hamper productivity.

Author Response

Please find the attached reply.

Reviewer 2 Report

Dear Authors, 

Your paper on the Zero Trust Model for Smart Manufacturing Industry is in line with the mission and methodologies of the target journal, and of the specific special issue.

However, some improvements are needed before being considered eligible for publication. Below are some indications: 

-Abstract. the abstract should clearly describe the aim of the article, its relevance and its methodology.  

- Introduction. the introduction mentions the concept of "digital ecosystems" but without provide a definition of ecosystems. The methodology and originality of the article as well as its contribution should be specified better. 

The main content. starting from paragraph n.2 there are several paragraphs, most of them are just a few lines, which makes it difficult to read, it is better to aggregate some parts. Some summarizing tables of the main concept could also be included to clarify the main topics. 

Methodology. it is not clear what is methodology adopted. Is it a general review followed by a proposal? more space should be given to the proposed solution from the abstract and introduction. Maybe a paragraph on methodology could help explain the methods of the article. 

Author Response

Please find the attached reply.

Round 2

Reviewer 2 Report

The Authors have improved their paper following the indications provided in the first round.